# Intergenerational effects of early adversity on survival in wild baboons

**Matthew N Zipple[1]\*, Elizabeth A Archie[2,3], Jenny Tung[1,3,4,5], Jeanne Altmann[3,6], Susan C Alberts[1,3,4,5]\***

[1]Department of Biology, Duke University, Durham, United States; [2]Department of Biological Sciences, University of Notre Dame, South Bend, United States; [3]Institute of Primate Research, National Museums of Kenya, Nairobi, Kenya; [4]Department of Evolutionary Anthropology, Duke University, Durham, United States; [5]Duke Population Research Institute, Duke University, Durham, United States; [6]Department of Ecology and Evolutionary Biology, Princeton University, Princeton, United States

**Abstract** Early life adversity can affect an individual's health, survival, and fertility for many years after the adverse experience. Whether early life adversity also imposes intergenerational effects on the exposed individual's offspring is not well understood. We fill this gap by leveraging prospective, longitudinal data on a wild, long-lived primate. We find that juveniles whose mothers experienced early life adversity exhibit high mortality before age 4, independent of the juvenile's own experience of early adversity. These juveniles often preceded their mothers in death by 1 to 2 years, indicating that high adversity females decline in their ability to raise offspring near the end of life. While we cannot exclude direct effects of a parent's environment on offspring quality (e.g., inherited epigenetic changes), our results are completely consistent with a classic parental effect, in which the environment experienced by a parent affects its future phenotype and therefore its offspring's phenotype.
DOI: https://doi.org/10.7554/eLife.47433.001

**\*For correspondence:**
matthew.zipple@duke.edu (MNZ);
alberts@duke.edu (SCA)

**Competing interests:** The authors declare that no competing interests exist.

## Introduction

An individual's health, survival, and fertility can be profoundly shaped by its early life environment (*Uller et al., 2013*). For example, in humans, low early life socioeconomic status predicts increased risk of mortality and many measures of poor health in adulthood (*Naess et al., 2004*; *Beebe-Dimmer et al., 2004*; *Kittleson et al., 2006*; *Smith et al., 1998*; *Frankel et al., 1999*; *Lidfeldt et al., 2007*; *van de Mheen et al., 1998*; *Kuh et al., 2002*; *Galobardes et al., 2004*). Similarly, several studies of wild mammals and birds (*Lea et al., 2015*; *Douhard et al., 2014*; *Nussey et al., 2007*; *Pigeon and Pelletier, 2018*; *Herfindal et al., 2015*; *Balbontín and Møller, 2015*; *Millon et al., 2011*) find that adult fecundity is reduced in animals that experienced adverse early life environments, and some have also found an effect of early life adversity on adult survival (*Nussey et al., 2007*; *Pigeon and Pelletier, 2018*; *Herfindal et al., 2015*; *Tung et al., 2016*).

If the effects of early adversity extend to the descendants of exposed individuals, the epidemiological and evolutionary impact of these effects would be further amplified. However, in humans, evidence that intergenerational effects stem directly from parental experience is mixed, as studies have produced somewhat contradictory results (*Veenendaal et al., 2013*; *Painter et al., 2008*; *Kaati et al., 2007*; *Pembrey et al., 2006*). For example, a study of the historical Överkalix population in Sweden identified strong, contrasting effects of grandparents' exposure to early-life food scarcity on grand-offspring survival, depending on small differences in the age at which the grandparent was exposed to scarcity (*Pembrey et al., 2006*). Similarly, two studies of a population that

**eLife digest** Experiences early in life can have lasting effects on the health and survival of humans and other creatures. Whether early hardships can also influence the wellbeing of the next generation is less clear. One previous study with captive hamsters suggested that adversity early in the life of a mother may indeed shorten how long her offspring will live. But hamsters only live for a few years and much less is known about the possibility for intergenerational effects in animals with longer lifespans. This is partly because such studies are time-consuming and thus more difficult to complete.

Over the past 45 years, scientists have collected data on generations of baboons living in and around the Amboseli National Park in southern Kenya. Baboons live in social groups with a strict hierarchy, and individuals can live for up to 30 years in the wild. Previous research has shown that early life adversity – such as being orphaned or simply having a low-ranking mother – can shorten the lifespan of female baboons even if they make it to adulthood. It was unclear, however, whether these ill effects could be passed on to the next generation. Now, Zipple et al. have used the wealth of data about the Amboseli baboons to find the answer.

After taking into account any adversity that each baboon experienced directly, Zipple et al. showed that juvenile baboons whose mothers were orphaned before reaching adulthood were 44% more likely to die young than juveniles whose grandmothers survived during their mother's early years. Baboons whose mothers had a close-in-age younger sibling were also 42% more likely to die early as compared to those whose mothers did not, perhaps because the younger sibling competed with the mother for access to maternal care.

The analysis suggests that early life adversity in female baboons can have intergenerational effects. More studies are needed to determine if this is also true of humans. If it is, such a result may help explain the persistence of poor health outcomes across generations and shed light on how best to intervene to interrupt this transmission.

DOI: https://doi.org/10.7554/eLife.47433.002

was exposed in utero to the Dutch hunger winter (a famine that resulted from a German blockade of the Netherlands during the winter of 1944–1945) found contradictory, sex-specific intergenerational effects, in one case suggesting an intergenerational effect that depended only upon the mother's early experience (*Painter et al., 2008*), and in the other case an effect that depended only upon the father's early experience (*Veenendaal et al., 2013*).

Compelling evidence for intergenerational effects of early adversity faced only in the parental generation comes from numerous laboratory studies of short-lived animals, which find strong relationships between a female's early life environment and the body size of her offspring (*Huck et al., 1986*; *Alonso-Alvarez et al., 2007*; *Helle et al., 2012*; *Goerlich et al., 2012*; *Taborsky, 2006*; *Beckerman et al., 2003*; *Saastamoinen et al., 2013*; *Vijendravarma et al., 2010*; *Fischer et al., 2003*; *Jobson et al., 2015*; reviewed in *Burton and Metcalfe (2014)*; but see *Bowers et al., 2017* for a rare example in the wild on house wrens). These findings provide important evidence that intergenerational effects of early adversity can occur. However, these studies do not address whether intergenerational effects of early adversity, independent of parent-offspring environmental correlations, occur in natural populations of long-lived animals. And while a few studies of short-lived captive animals have demonstrated a relationship between a female's early environment and her offspring's survival or reproduction (*Huck et al., 1987*; *Naguib et al., 2006*; *Marcil-Ferland et al., 2013*), the ecological validity of these findings has yet to be verified by studying intergenerational fitness effects in a population of wild and/or long-lived animals. In wild populations, animals are exposed only to natural, unmanipulated levels of early adversity, and are also subject to any social factors which might mitigate or aggravate the influence of those early adverse events.

Addressing whether the effects of early adversity in one generation affect reproduction or survival in the next is challenging because of the difficulties of linking high-quality data on early adversity in one generation to health and survival outcomes in the next. Here, we overcome these challenges by taking advantage of a prospective longitudinal dataset from a natural primate population: the baboons of the Amboseli ecosystem in southern Kenya (*Alberts and Altmann, 2012*). This dataset

includes 45 years of individual-based data on early adversity, and real-time observations of later-life survival outcomes for hundreds of subjects with known maternities and grand maternities. Moreover, unlike many human populations, we do not observe inter-generational transmission of adverse conditions; that is, offspring of females who experienced early life adversity are not more likely to experience early life adversity themselves (except in the case of inheritance of low social rank, see below), allowing us to avoid this common confound in human societies.

To test for intergenerational effects of early adversity, we focused on early adversity experienced by female baboons who later became mothers, and whose offspring were also in our dataset. We asked whether the early adversity experienced by these females ('maternal early adversity') predicted the survival of their juvenile offspring in the next generation, after controlling for the early adversity directly experienced by the offspring themselves.

We considered five types of early adverse conditions (*Table 1*), based on previous work in our study population that demonstrated effects of these conditions on a female baboon's own adult survival (*Tung et al., 2016*). These included: (i) maternal death during development (0–4 years of age), which indicates the loss of an important source of social support, physical protection, and nutrition (*Altmann, 1980*; *Lea et al., 2014*), (ii) being born to a low-ranking mother, which influences growth rates and age at maturation (*Charpentier et al., 2008*; *Altmann and Alberts, 2003a*; *Altmann et al., 1988*) (iii) being born into a large social group (and thus experiencing high density conditions and high levels of within-group competition) (*Lea et al., 2015*; *Charpentier et al., 2008*; *Altmann and Alberts, 2003b*) (iv) being born during a drought, which reduces fertility in adulthood (*Lea et al., 2015*; *Beehner et al., 2006*), and (v) experiencing the birth of a close-in-age younger sibling, which may reduce maternal investment received during development (*Altmann et al., 1978*). Importantly—and in contrast to research on humans (*Felitti et al., 1998*)—sources of early adversity are not strongly correlated in our population, which allows us to measure the independent effects of different sources of adversity (*Supplementary file 1* Table S1).

## Results

We built a mixed effects Cox proportional hazards model of offspring survival during the juvenile period that included early adversity measures present in the mother's and the offspring's early life as

**Table 1.** Early adverse conditions and the frequencies with which they occur in maternal and offspring generations of our dataset.

| Adverse Condition[*] | Criterion | Frequency | |
|---|---|---|---|
| | | *Maternal Generation* | *Offspring Generation* |
| Drought | During the first year of life, the focal individual experienced less than 200 mm of rainfall (i.e., drought conditions; *Beehner et al., 2006*). | 0.09 | 0.15 |
| High Social Density | The individual was born into a group with a high social density (>35 adults), indicating high levels of within-group competition. | 0.06 | 0.32 |
| Maternal Loss | The mother of the focal individual died within four years of the individual's birth. | 0.21 | 0.25 |
| Low Maternal Rank[†] | The focal individual was born to a mother with a low social rank (mother's rank fell in the bottom quartile of the group's dominance hierarchy, rank < 0.25). | 0.17 | 0.23 |
| Close-In-Age Younger Sibling[‡] | The focal individual had a younger sibling born to its mother within 18 months of the focal's birth. | 0.20 | – |

[*]These criteria were used in a previous analysis in our population (*Tung et al., 2016*), with the exception of maternal rank, which is evaluated here as a proportional measure rather than an ordinal one as in the previous analysis.

[†]Proportional rank is the proportion of other adult females in a group that an individual's mother outranks. The reduced frequency with which low maternal rank appears in the maternal generation is a likely a result of offspring of low-ranking mothers surviving less well (*Silk et al., 2003*), and therefore not surviving to appear as mothers in our dataset.

[‡]We excluded the birth of a close-in-age younger sibling for the offspring generation to avoid including a potential reverse-causal factor in our model: the closest-in-age siblings in our dataset occur as a result of the focal offspring's death, because female baboons (who are not seasonal reproducers) accelerate their next conception after the death of a dependent offspring.

DOI: https://doi.org/10.7554/eLife.47433.003

binary fixed effects. We defined the juvenile period based on survival until age 4, near the age of menarche for females and earliest dispersal for males in this population (*Charpentier et al., 2008*). We included data on maternal early adversity for all five adverse early life conditions, and we included data on offspring early adversity for four of the five conditions. We excluded the birth of a close-in-age younger sibling for the offspring generation to avoid including a potential reverse-causal factor in our model. Specifically, the closest-in-age siblings in our dataset tend to occur *as a result of* the focal offspring's death, because female baboons (who are not seasonal reproducers) quickly conceive again if they lose a dependent offspring. In these cases, early mortality would be correlated with short interbirth intervals, but only because the offspring's death predicts subsequent production of another infant, not because short interbirth intervals constitute a form of adversity. We included maternal and grandmaternal ID as random effects. In total, we used data collected from 1976 to 2017 to analyze the survival of 687 offspring (46.5% males) born to 169 females (mean 4.1 offspring per female, range 1–12) for whom we had data on all five adverse conditions in the mother's early life, and all four adverse conditions in the offspring's early life.

Each adverse condition was scored as present or absent for each subject, and each one affected a minority of our study subjects (range 6–34%). Mothers and offspring had similar chances of experiencing adverse conditions, except for social density: offspring were more likely than mothers to be born into large social groups because of population growth over the 5-decade study period (*Table 1*). Unlike typical patterns of early adversity in human populations (*Felitti et al., 1998*), different sources of early life adversity in our population were not strongly correlated with each other: no adverse condition explained more than 4% of the variance in any other condition, either within or between generations, with the exception of maternal rank in the mother and offsprng's generation (p<0.0001, $r^2$ = 0.16) (*Supplementary file 1* Table S1).

## Maternal Early Life Adversity and Offspring Survival

Our full multivariate Cox proportional hazards model for offspring survival (*Supplementary file 1* Table S2) included all nine early adverse conditions (five for mothers and four for offspring). We found strong negative effects of two characteristics of the *mother's* early life environment on their offspring's survival during the first 4 years of life: maternal loss (hazard ratio = 1.48, p=0.006) and presence of a close-in-age younger sibling (HR = 1.39, p=0.03). Following backwards model selection (performed by removing the parameters with the highest p values until only predictors with a p-value<0.05 remained), these two characteristics remained the only significant maternal early life predictors of offspring survival (*Table 2*, *Figure 1*, along with two conditions in the offspring's early life environment: see below). Adding maternal age, offspring sex or interactions between maternal age or offspring sex and sources of maternal adversity did not improve the fit of the model (*Supplementary file 1* Tables S3-S5).

In sum, offspring whose mothers experienced early maternal loss experienced a 48% higher probability of dying throughout the first four years of life than unaffected offspring, and offspring whose mothers had a close-in-age sibling experienced a 39% higher probability of dying than unaffected

**Table 2.** Reduced model of the effects of maternal and offspring early adversity on offspring survival during early life ($R^2$ = 0.07).

| Generation | Parameter[*] | Coefficient | Hazard ratio (95% CI) | P value | Interpretation |
|---|---|---|---|---|---|
| *Maternal* | Maternal Loss | 0.37 | 1.44 (1.10–1.90) | 0.009 | Offspring survive less well if their mother experienced maternal loss during her early life. |
| | Close-in-age Younger Sibling | 0.35 | 1.42 (1.06–1.90) | 0.018 | Offspring survive less well if their mother had a close-in-age younger sibling during her early life. |
| *Offspring* | Maternal Loss | 0.68 | 1.98 (1.53–2.56) | $3 \times 10^{-7}$ | Offspring survive less well if they experienced maternal loss within four years of their birth. |
| | Low Maternal Rank | 0.43 | 1.54 (1.17–2.01) | 0.002 | Offspring survive less well if they were born to a low-ranking mother. |

[*]An alternative model that considered cumulative maternal adversity was not a better or worse fit than the reduced multivariate maternal adversity model (see *Supplementary file 1* Table S6. For both the model presented here and that in *Supplementary file 1* Table S6, $R^2$ = 0.07, log likelihood = −1598).
DOI: https://doi.org/10.7554/eLife.47433.005

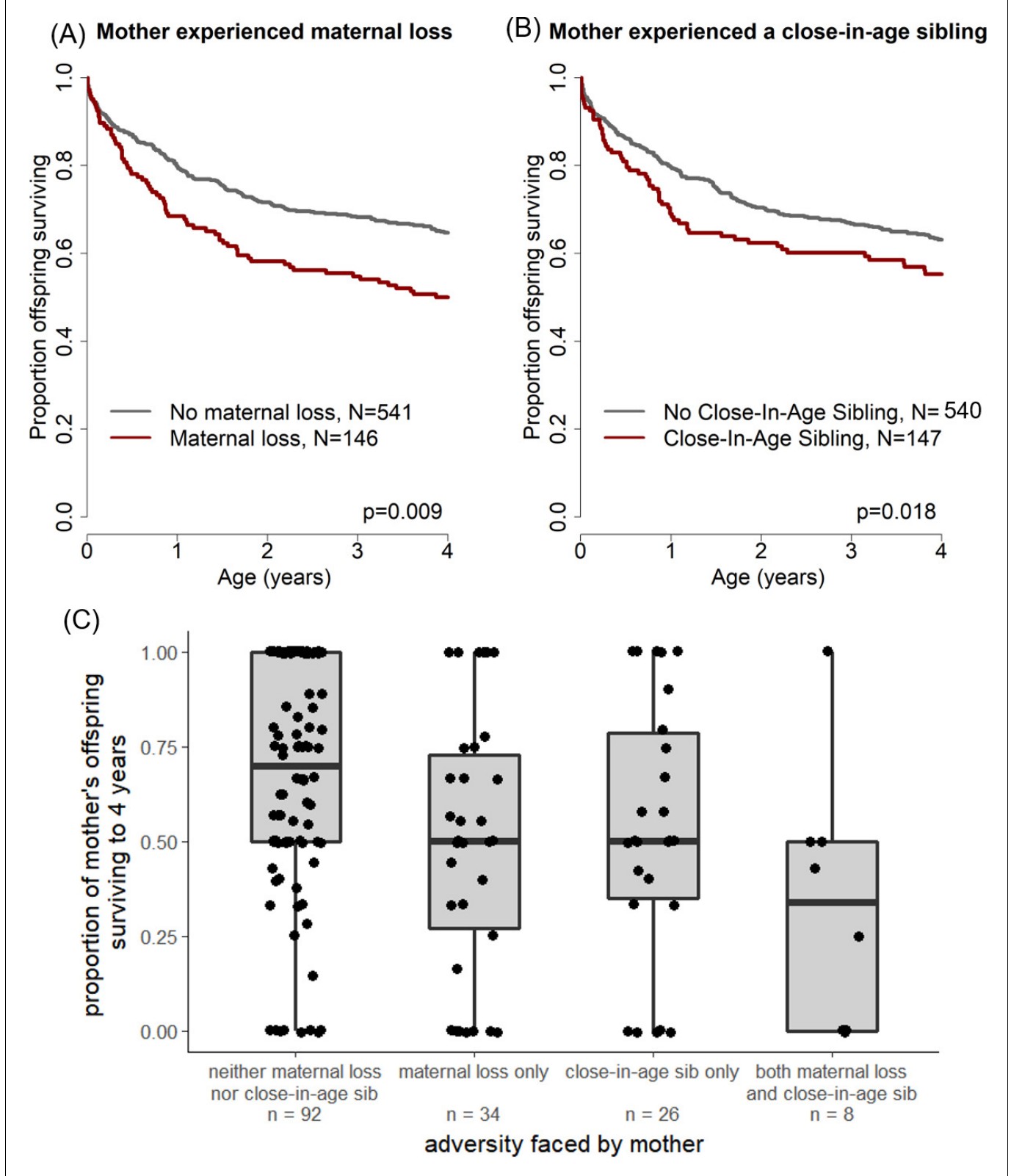

**Figure 1.** Offspring survival was influenced by characteristics of their mothers' early-life environments. Offspring survived relatively less well during the juvenile period if (**A**) their mother lost her own mother during her early life and/or (**B**) their mother experienced a close-in-age younger sibling. An alternative visualization of the data (**C**) shows an equivalent pattern when mothers, rather than offspring, are treated as the unit of analysis.
DOI: https://doi.org/10.7554/eLife.47433.004

offspring. This effect is striking especially considering that a median of 7.0 and 8.0 years separated the offspring's own birth from the mother's experience of maternal loss or birth of a close-in-age sibling, respectively. A similar pattern holds if mothers, rather than offspring, are treated as the unit of analysis: mothers who experienced early adversity have lower average offspring survival than mothers who did not (see *Figure 1c*).

Notably, previous work in our population found that these two sources of adversity—maternal loss and the presence of a close-in-age younger sibling during early life—are also sources of mortality risk once females reach adulthood, and in fact are the two strongest predictors of adult survival among six different early-life conditions considered (*Tung et al., 2016*). Hence, early-life conditions that are especially adverse for females when they reach adulthood also negatively affect the survival of their offspring.

Both the full and reduced models of offspring survival also included two conditions in the *offspring's* early life environment as significant predictors of juvenile survival. Specifically, maternal loss experienced by the offspring and low maternal rank during the offspring's juvenile period had strong negative effects on offspring survival (*Supplementary file 1* Table S2, maternal death: Hazard Ratio = 1.95 [1.51–2.54], $p=5\times10^{-7}$, low maternal rank: Hazard Ratio = 1.43 [1.05–1.94], p=0.025). Thus, maternal loss in the offspring's generation had a stronger effect on offspring survival (nearly doubling offspring mortality risk) than maternal loss in the mother's generation. In contrast, the effect of having a low-ranking mother, which was associated with a 43% increase in offspring mortality risk, was comparable in its effect size to the two significant predictors from the maternal generation (maternal loss and close-in-age sibling for the mother, 48% and 39% increase in offspring mortality, respectively). Thus, two adverse conditions in a mother's early life had as large or larger of an impact on her offspring's survival than all but one adverse condition experienced by the offspring directly.

## Maternal Viability and Offspring Survival

The strong effect of the mother's death on offspring survival prior to four years (*Table 2*) is unsurprising at first consideration: the most obvious explanation for this effect is that offspring depend upon their mothers, so that if the mother dies the offspring is also likely to die at the same time or die subsequently. Indeed, this sequence of events does occur in our population: of the 32 offspring that were alive and less than eight months old when their mother died, 31 (97%) died before reaching one year of age.

However, offspring death could also precede maternal death if it acts as a *harbinger* of the mother's death, as opposed to a *consequence* of it. In this scenario, offspring mortality risk is increased because their mothers are in poor condition and hence unable to provide adequate care or resources to the offspring. This hypothesis therefore proposes an alternative causal chain from poor maternal health to offspring death, which would occur while the mother is still alive.

To examine whether this phenomenon occurs in our study population, we modeled offspring survival to age 2 years (halfway through the juvenile period) as a function of maternal death during years 2–4 after an offspring's birth (i.e., the two years that *followed* the offspring survival period modeled in the response variable). In this analysis, we considered only the subset of offspring in our dataset whose mothers survived the entire first two years of the offspring's life, and for whom we were able to evaluate the four significant predictors of offspring survival identified above and in *Table 2* (N = 671). Our results were striking: offspring were less likely to survive during the first two years of life if their mothers died 2–4 years after their birth. In other words, these offspring were more likely to die even when their mother was still present (hazard ratio = 1.50 [1.01–2.23], p=0.045).

To test for a role of maternal early adversity in this effect, we next partitioned our analysis of offspring survival to age two based on whether the mother experienced either maternal loss or a close-in-age younger sibling (i.e., either or both of the two maternal early life conditions that significantly predicted their offspring's survival; *Table 2*). We found that, among offspring whose mothers experienced either or both of these two adverse events (N = 247), maternal death in years 2–4 after the offspring's birth significantly predicted reduced offspring survival to age 2 years (*Figure 2a*, hazards ratio = 1.78, 95% CI = [1.05–3.01], p=0.034). Maternal death in the same period did not, however, predict reduced offspring survival when mothers had not experienced maternal loss or a close-in-age younger sibling (N = 424; *Figure 2b*, hazard ratio = 1.21, 95% CI = [0.7–2.2], p=0.53). Hence,

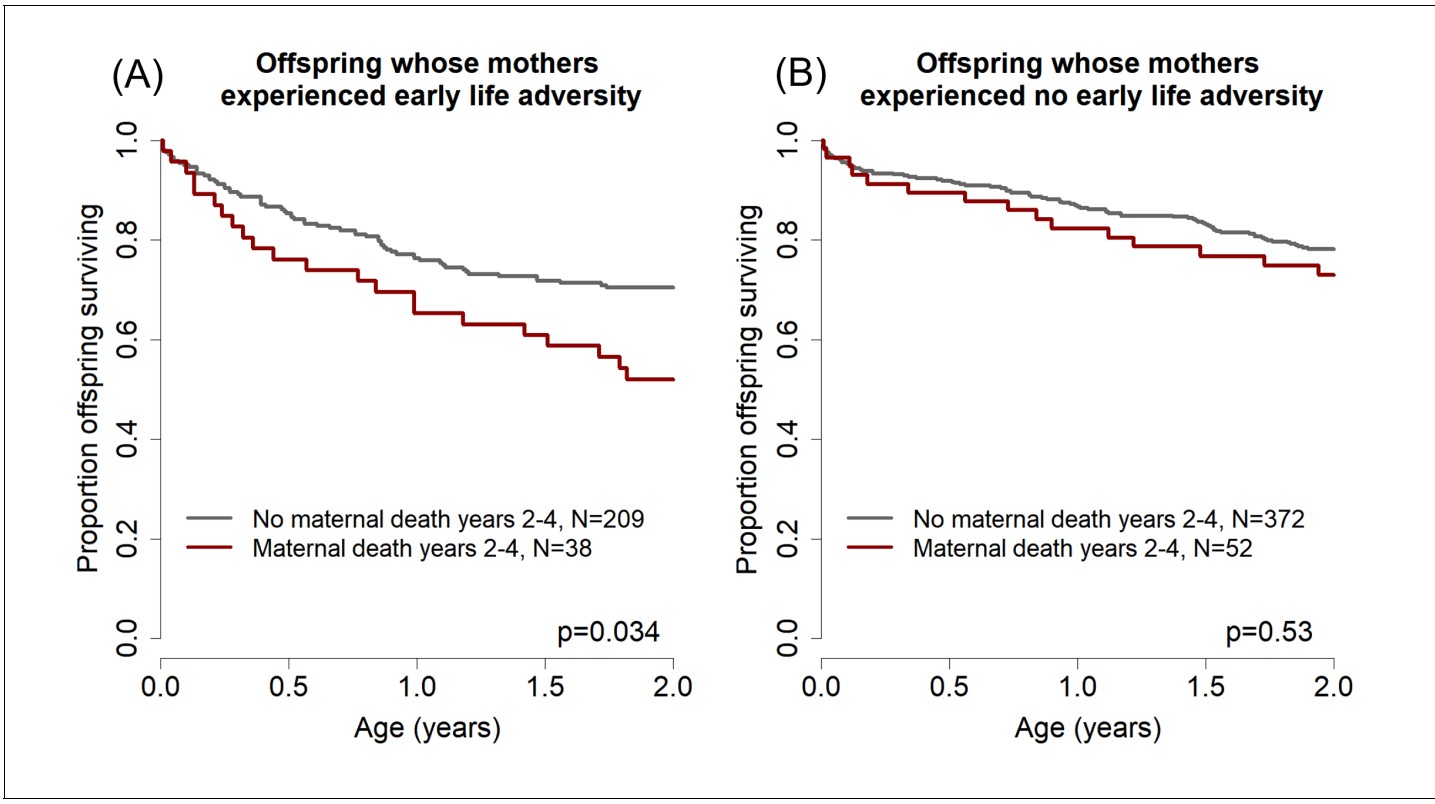

**Figure 2.** Effects of maternal adversity on offspring survival are explained by reduced maternal viability. (**A**) Among those offspring whose mothers experienced significant early life adversity (maternal loss and/or a competing younger sibling), poor offspring survival from ages 0–2 (while the mother was still alive) was predicted by maternal death in years 2–4 after the offspring's birth. (**B**) In contrast, there was no relationship between offspring survival in the first two years of life and maternal death in years 2–4 for the offspring of mothers who did not experience early life adversity.
DOI: https://doi.org/10.7554/eLife.47433.006

the pattern we observed when analyzing the full data set of offspring that survived to age 2 (N = 671) is completely driven by the offspring of mothers who experienced substantial early adversity. This finding is consistent with the hypothesis that maternal early life adversity results in compromised maternal condition in adulthood, which in turn results in both earlier death for adult females and a reduction in their ability to successfully raise offspring towards the end of their lives (i.e., a maternal effect on the offspring generation).

## Discussion

We have demonstrated that adverse environmental conditions during the early life of a female baboon, which are already known to negatively affect both her survival (*Tung et al., 2016*) and her reproduction (*Lea et al., 2015*) in adulthood, also reduce the survival of her offspring. Importantly, this effect is independent of the environment experienced by those offspring themselves (*Figure 1*). The reduction in offspring survival is likely linked to reductions in maternal viability: mothers that experienced early life adversity are significantly less able to successfully raise offspring born near the ends of their lives, while the same is not true for mothers that did not experience early life adversity (*Figure 2*). Together, these findings support the hypothesis that early life adversity produces constraints during development that lead not only to reduced adult survival and lifetime reproductive success (*Tung et al., 2016*) but also to a reduced ability to successfully raise those offspring that are produced (*Figure 2a*). We did not identify any sex-specific intergenerational effects of maternal early adversity.

The results reported here help to fill a key gap in the literature concerning the intergenerational effects of early life adversity on survival. Human studies have yielded inconsistent results on this topic

thus far when maternal and offspring environments are not correlated: different studies on the same populations have reported contradictory sex-specific effects on health (*Veenendaal et al., 2013*; *Painter et al., 2008*) or have found that small differences in the age at which subjects' parents or grandparents were exposed to adversity can lead to a reversal in the direction of these effects (*Kaati et al., 2007*; *Pembrey et al., 2006*). Among studies in non-human animals, several studies in fish (*Donelson et al., 2008*; *Venturelli et al., 2010*), reptiles (*Warner and Lovern, 2014*), birds (*Blomqvist et al., 1997*; *Ridley, 2007*), and ungulates (*Cameron et al., 1987*; *Théoret-Gosselin et al., 2015*; *Keech et al., 2000*; *Clutton-Brock et al., 1987*; *Clutton-Brock et al., 1984*) have found that parental body condition at the time of offspring birth influences offspring survival, but none have linked parents' early adverse experiences to offspring survival. Additionally, while previous studies have identified effects of parental early adversity on offspring traits in a limited number of captive, short-lived systems (*Burton and Metcalfe, 2014*; *Huck et al., 1987*; *Naguib et al., 2006*), ours is the first to link parental early adversity to an important component of offspring fitness in a wild, long-lived animal.

Our findings help to explain the persistence of health deficits across generations (*Aizer and Currie, 2014*; *Kane et al., 2018*; *Cnattingius et al., 2012*), by revealing that in long-lived primates, the early life experiences of mothers have important implications for offspring health and survival. Recent studies in humans have demonstrated that conditions experienced by mothers during pregnancy (e.g., low SES, psychosocial stress, mood dysregulation, prenatal smoking) can affect HPA axis regulation (*Thayer and Kuzawa, 2014*; *Entringer et al., 2009*) and birthweight (*Aizer and Currie, 2014*; *Kane et al., 2018*) in her offspring. These and other maternal characteristics present during pregnancy are influenced not only by mothers' experiences in adulthood, but also by the long-term effects of environmental conditions experienced in mothers' early lives (*Kane et al., 2018*; *Kuzawa, 2005*). Our findings therefore motivate future work to test for comparable intergenerational fitness effects of early adversity in humans and other non-human animals.

Our findings are consistent with the hypothesis that early adversity results in intergenerational developmental constraints (*Lea et al., 2015*; *Grafen, 1988*; *Monaghan, 2008*; *Lea et al., 2017a*) and are not consistent with an intergenerational predictive adaptive response hypothesis (*Monaghan, 2008*; *Gluckman et al., 2005*; *Herman et al., 2014*). Rather than being buffered against the effects of maternal loss, those offspring that experienced maternal loss and whose mothers had also experienced maternal loss were more likely, not less likely, to die, as compared to offspring that experienced maternal loss but whose mothers did not. Thus, individuals in the offspring generation experience constraints not only as a result of their own early environment, but also as a result of their mothers' developmental histories, including events that occurred years before their own conception. Our results are consistent with the hypothesis that a female's condition at the time of her offspring's conception and/or birth reflects her previous experiences, and that her condition thereby influences the development and survival of her offspring (*Kuzawa, 2005*; *Kuzawa, 2017*; *Lea et al., 2017b*).

Our study is unable to definitively identify the mechanism by which effects of early adversity may be transmitted from parent to offspring. However, our finding that reduced offspring survival appears to be partially mediated by reduced maternal viability suggests that the mode of transmission is most readily explained as a classic parental effect, in which early life adversity affects the phenotypic quality of the mother during adulthood, and in turn affects her offspring's development (*Mousseau and Fox, 1998a*; *Mousseau and Fox, 1998b*; *Russell and Lummaa, 2009*; *Badyaev and Uller, 2009*). Recently, intergenerational transmission of adversity has been discussed as a potential consequence of inherited epigenetic changes (*Heard and Martienssen, 2014*). While we cannot exclude this possibility, our results are a reminder that simpler mechanisms—in this case, a classic maternal effect—may be a more parsimonious (albeit non-mutually exclusive) explanation.

Notably, the importance of both maternal death and a close-in-age younger sibling suggest that maternal investment may be key to understanding the intergenerational developmental constraints we observed. Both maternal death and the presence of a close-in-age sibling suggest a possible reduction in the amount of maternal investment that the mothers in our analysis received during their early life. Maternal loss, even after weaning, may affect a developing primate's ability to learn to forage, to avoid social harassment, and to receive social benefits, such as grooming, that are linked to health (*King, 1994*; *Janson and van Schaik, 1993*; *Akinyi et al., 2013*; *Walters, 1987*; *Altmann, 1998*; *Ezenwa et al., 2016*). Having a close-in-age sibling likely predicts a relatively early weaning event, which may reflect less maternal provisioning than would occur with more delayed

weaning and a longer birth interval (*Hinde and Milligan, 2011*; *Mattison et al., 2015*; *Silk, 1988*). Thus, we hypothesize that mothers who lost their own mothers or had a close-in-age sibling suffered reduced energetic and social input from their mothers, which subsequently led to lifelong developmental constraints. Additionally, these females may not have had adequate time to learn from their mothers how to provide high quality maternal care later in life. While we do not routinely collect detailed data on maternal care as part of long-term monitoring, the results reported here motivate targeted analyses of how maternal adversity relates to differences in maternal care and style in future work.

## Materials and methods

### Study system

The Amboseli Baboon Research Project is a long-term longitudinal study of wild baboons living in and around Amboseli National Park, Kenya. A detailed description of the study system can be found elsewhere (*Alberts and Altmann, 2012*). Researchers have continuously collected behavioral, environmental, and demographic data from the population since 1971. All subjects are visually recognized, and near-daily censuses allow us to precisely document the timing of demographic events, including the birth and death of study individuals. Critical to this study, we have continuously collected near-daily measures of group size, daily rainfall levels (beginning in 1976), and monthly calculations of social dominance rank (*Hausfater, 1975*).

### Study Subjects

In our analyses of offspring survival, we included all individuals who met two criteria: (i) they lived in social groups that fed exclusively on wild foods rather than having their diet supplemented with human-sourced refuse; and (ii) we were able to evaluate each of the five sources of maternal early life adversity and four sources of offspring early life adversity outlined below. Although transmission of paternal early adversity may also occur in our population, we did not consider it here because we knew paternal identities for only a subset of our study subjects and had early life data on only a subset number of fathers. Our analysis ultimately relied on data spanning more than four decades, from 1976 to 2017.

### Measuring Early Life Adversity

Previous work in the Amboseli population defined six binary indicators of early life adversity and considered a single index of cumulative adversity based on the sum of these indicators (*Tung et al., 2016*). This cumulative adversity index is a strong predictor of adult lifespan: females that experienced high levels of early life adversity (i.e., a greater number of adverse early life conditions) but still survived to adulthood lived dramatically shorter lives compared to females that did not experience early adversity (*Tung et al., 2016*). In addition to the five sources of early adversity discussed above, this previous analysis also considered early social connectedness (social integration versus social isolation) as a sixth source of adversity (*Tung et al., 2016*). Social connectedness data are missing for some mothers who were born relatively early in the long-term study. To maximize our sample size, we therefore did not include measures of social connectedness in this analysis.

Our operational definitions for each source of adversity mirrored those used by *Tung et al. (2016)* for the remaining five conditions, except that here we employed measures of proportional rather than ordinal dominance rank (i.e., rank measured as a proportion of females that the focal individual dominates, rather than her ordinal rank number). We also built an index of cumulative maternal adversity, but because that model did not fit the data better than our reduced multivariate model (in contrast to the results for adult female survival; *Tung et al., 2016*) we report the multivariate model in the main text. The alternative model based on cumulative maternal adversity is presented in *Supplementary file 1* Table S6.

### Statistical Analysis

We built a mixed effects Cox proportional hazards model of offspring survival during the first four years of life using the R package coxme (*Therneau, 2012*; *R Development Core Team, 2018*). The response variable in our model was the age at which offspring death occurred (if at all) during the

first 4 years of life. We considered offspring survival to age four as the key survival period of interest because it roughly corresponds to the end of the juvenile period for baboons (*Charpentier et al., 2008*). Offspring that survived beyond age four were treated as censored individuals who survived until at least age 4. In our models of offspring survival as a function of maternal viability (*Figure 2*), we altered the first model to predict survival during the first two years of life as a function of maternal survival during years 2–4 after offspring birth (see *Supplementary file 1* Table S7 for model syntax).

## Data availability

Datasets presented in this article can be downloaded from Dryad using the following Digital object identifier (DOI): 10.5061/dryad.4hc8k1r (*Zipple et al., 2019*).

## Additional information

### Funding

| Funder | Grant reference number | Author |
|---|---|---|
| National Institute on Aging | R01-AG053308 | Susan C Alberts |
| National Institute on Aging | P01-AG031719 | Susan C Alberts |
| Leakey Foundation | | Matthew N Zipple |
| National Science Foundation | BCS 1826215 | Matthew N Zipple<br>Susan C Alberts |
| Duke University | | Matthew N Zipple<br>Jenny Tung<br>Susan C Alberts |
| Princeton University | | Jeanne Altmann |
| University of Notre Dame | | Elizabeth A Archie |
| National Science Foundation | IOS 1456832 | Susan C Alberts |
| National Institute on Aging | R01-AG053330 | Elizabeth A Archie |
| Eunice Kennedy Shriver National Institute of Child Health and Human Development | R01-HD088558 | Jenny Tung |

The funders had no role in study design, data collection and interpretation, or the decision to submit the work for publication.

### Author contributions

Matthew N Zipple, Conceptualization, Formal analysis, Visualization, Methodology, Writing—original draft, Writing—review and editing; Elizabeth A Archie, Jenny Tung, Jeanne Altmann, Resources, Data curation, Funding acquisition, Investigation, Methodology, Project administration, Writing—review and editing; Susan C Alberts, Conceptualization, Resources, Data curation, Supervision, Funding acquisition, Investigation, Methodology, Writing—original draft, Project administration, Writing—review and editing

### Author ORCIDs

Matthew N Zipple  https://orcid.org/0000-0003-3451-2103
Jenny Tung  http://orcid.org/0000-0003-0416-2958
Susan C Alberts  https://orcid.org/0000-0002-1313-488X

### Decision letter and Author response

Decision letter https://doi.org/10.7554/eLife.47433.012
Author response https://doi.org/10.7554/eLife.47433.013

## Additional files

### Supplementary files

• Supplementary file 1. Tables S1-S7.
DOI: https://doi.org/10.7554/eLife.47433.007

• Transparent reporting form
DOI: https://doi.org/10.7554/eLife.47433.008

### Data availability

All data used in analyses have been anonymized and made available on Dryad (https://doi.org/10.5061/dryad.4hc8k1r).

The following dataset was generated:

| Author(s) | Year | Dataset title | Dataset URL | Database and Identifier |
|---|---|---|---|---|
| Zipple MN, Archie EA, Tung J, Altmann J, Alberts SC | 2019 | Data from: Intergenerational effects of early adversity on survival in wild baboons | https://doi.org/10.5061/dryad.4hc8k1r | Dryad Digitial Repository, 10.5061/dryad.4hc8k1r |

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
