## [Decision Letter]

Thank you for submitting your article "Intergenerational effects of early adversity on survival in wild baboons" for consideration by *eLife*. Your article has been reviewed by three peer reviewers and overseen by Eduardo Franco as the Senior Editor, Molly Przeworski as Reviewing Editor and three reviewers. The following individuals involved in review of your submission have agreed to reveal their identity: Zane Thayer (Reviewer #1); Daniel Blumstein (Reviewer #2); Chris Kuzawa (Reviewer #3).

As you can see, all three reviewers agree that this work is an interesting and important contribution, but suggest a few additional analyses and ask for clarifications and revisions to the text. Reviewer #3 also raises a more substantial concern about how the paper is framed, in particular in relation to the related work in humans. We include the detailed comments below to help guide you with the revisions.

*Reviewer #1:*

This manuscript utilizes a large, longitudinal dataset on wild living baboons to investigate the relationship between early adversity experience and survival of offspring. The question is important, the data are very valuable, and the general approach is sound. That said, revisions that could further improve the quality of the manuscript.

1) Given that there are a mean of 4.1 offspring per female, I would have liked to see a simple analysis that looked at overall fitness for mothers (% that survived to adulthood) who experienced early adversity vs. those that did not. In addition, I am wondering if parity is important, as based on the analysis in subsection “Maternal Viability and Offspring Survival” it would imply that it is. Is the relationship between parity and offspring survivorship different depending on maternal adversity experience? This is implied by the analysis described subsection “Maternal Viability and Offspring Survival” but is not actually assessed in that analysis.

2) The logic of why short inter-birth interval is used to as a measure of early maternal adversity, but not offspring adversity, is not clear to me.

3) I do not find the analysis with the 10-minute focal follows (subsection “Maternal Early Life Adversity and Quantity of Maternal Care”) to be very helpful, as it seems unlikely that much useful information about the hypothesis can be gleaned from that short of a time. Using such data to assess how early adversity could shape maternal care is, however, a good idea, and instead of presenting this analysis I would recommend instead including a paragraph about this hypothesis/suggested method in the Discussion section as an important area of future research.

4) In results the authors critique inconsistent sex-specific effects in human literature (Discussion section) – but do the authors look for sex-specific effects in their analysis? Given prior research that suggests sex-differences in sensitivity to early environmental experience this seems like something important to assess.

5) I would have appreciated seeing more of a discussion of why the authors believe the variables that came out as significant – in particular maternal death and a short interbirth interval – are most strongly associated with offspring survival.

*Reviewer #2:*

This is a fascinating study that shows that early adverse experiences may have profound effects on future health and survival of free-living baboons in an exceptionally well-studied system – the long-term Amboseli study. What's neat is that not all types of adverse experiences have equally profound costs and the authors identify those losses that are most costly to young baboons. Importantly, these effects are seen transgenerationally – the offspring of deprived mothers also have detectable deficits – a depressing but important result that changes how we should consider experiences in wildlife (and potentially humans). But there's more, the authors were able to specifically reject a predictive adaptive response (PAR) model (an important life history model with mixed support in other systems) which predicted that offspring of deprived mothers would do better, not worse, if they too faced early life adversity. They don't. Bad experiences are costly to ones' own life prospects and ones' offspring's life prospects.

This is an important paper for all of the reasons the authors articulated – especially the identification of inter/transgenerational effects.

The statistics were generally well described. The paper was exceptionally well-written. Materials and methods: I much prefer the full multivariate approach rather than a unit-weighted approach (described in the Materials and methods section – which has all sorts of assumptions about the relative costs of different experiences). I'm glad these multivariate results were presented.

*Reviewer #3:*

This is an important study demonstrating a likely intergenerational maternal effect on juvenile survival among baboons at Amboseli, one of the longest running studies of its kind. The authors use multiple decades of data and a large sample to report evidence that a female's early life stress experiences can influence survival of their unexposed juvenile offspring, independent of their own early life experiences. This is an important finding that extends study of multi-generational parental effects to a long-lived mammal/primate. It also provides evidence against the applicability of the predictive adaptive response hypothesis in this setting. Overall, the paper is nicely written and clear, and I have relatively minor suggestions on how to improve it.

The biggest issue that I see is in the set up and framing with respect to past literatures – the importance of the study is framed with respect to the human chronic disease-motivated DOHaD literature, which seeks to understand the developmental contributions to late life outcomes like diabetes or CVD. This study, in contrast, focuses on juvenile mortality. The authors do not provide adequate information on the most typical causes of juvenile death. And in any case, it is hard to see juvenile mortality (which I assume ultimately traces back to factors like resource access, social capital, violence?) as a model or analogue for the types of pathways and outcomes studied by DOHaD.

Which makes me wonder whether it might be more appropriate, and more to the point, to frame the primary contribution and motivation of this paper with respect to the parental effects literature, rather than DOHaD. I see little similarity (biologically) between what is being studied here and what the DOHaD literature studies, so it feels a bit distracting to have DOHaD as the opening framing – especially given that the authors never come back to that literature or discuss the limited insights that these findings provide into it. Another possible, generic framing is a test of the PAR idea – these findings showing pretty clear evidence against its applicability in this case.

---

## [Author Response]

Reviewer #1:This manuscript utilizes a large, longitudinal dataset on wild living baboons to investigate the relationship between early adversity experience and survival of offspring. The question is important, the data are very valuable, and the general approach is sound. That said, revisions that could further improve the quality of the manuscript.1) Given that there are a mean of 4.1 offspring per female, I would have liked to see a simple analysis that looked at overall fitness for mothers (% that survived to adulthood) who experienced early adversity vs. those that did not.

We have added an additional visualization (now Figure 1C) that treats the mother, rather than her offspring, as the unit of analysis. We think that this figure provides a nice complement to Figures 1A and 1B and we thank the reviewer for suggesting it.

From the main text:

“Figure 1. Offspring survival was influenced by characteristics of their mothers’ early-life environments…An alternative visualization of the data (C) shows an equivalent pattern when mothers, rather than offspring, are treated as the unit of analysis.”

Although we have not included an additional formal analysis in the main text, we provide one below and are willing to add it to the supplement at the reviewer and editor’s discretion:

We built a linear model where the proportion of each female’s offspring that survived to age four is modeled as a function of the two components of maternal early adversity that significantly predicted offspring survival (maternal loss and presence of a close-in-age sibling). Such an analysis treats the mother, not the offspring as the unit of analysis. The results of this analysis mirror the analysis presented in the main text: mothers who experienced early maternal loss, a close-in-age younger sibling, or both have reduced offspring survival. These effects are independent and additive: effect of maternal loss in mother’s generation = -0.17, se = 0.06, t = -2.85, p = 0.005, d.f. = 157; effect of close-in-age sibling in mother’s generation = -0.13, se = 0.06, t = -2.1, p = 0.03; d.f. = 157.

In addition, I am wondering if parity is important, as based on the analysis in subsection “Maternal Viability and Offspring Survival” it would imply that it is. Is the relationship between parity and offspring survivorship different depending on maternal adversity experience? This is implied by the analysis described subsection “Maternal Viability and Offspring Survival” but is not actually assessed in that analysis.

We interpret the reviewer’s comment to specifically apply to the analysis of offspring survival to age 2, so we first discuss the importance of maternal parity in that analysis in detail, before also discussing the importance of parity for offspring survival to age 4.

The reviewer’s comment about a relationship between parity and offspring survival is an interesting idea, but not one that we intended to imply in our original analysis. Based on our previous findings, parity affects offspring survival in two ways in the Amboseli baboons: (i) primiparous mothers have marginally (p = 0.056) lower offspring survival than multiparous mothers (McLean et al., in press, https://doi.org/10.1086/705810); and (ii) females senesce in their ability to promote offspring survival, such that older mothers have lower offspring survival (McLean et al., in press, https://doi.org/10.1086/705810).

Because parity is tightly correlated with maternal age in our population (r^2^ = 0.86), we therefore tested whether maternal senescence could better account for the effects of impending maternal death on offspring survival to age 2. Adding maternal age as an additional fixed effect in the models of offspring survival to age 2 did not improve our models (p=0.64 for maternal age among mothers that experienced early adversity and p = 0.43 among those that did not).

We have provided the outputs of these models below. Although we have not included them in the supplement at this time, we will do so if the reviewer believes them to be useful.

Model output of a mixed effects model of offspring survival to age 2, where mothers experienced early maternal loss or the presence of a close in age sibling:

ParameterHazard Ratio Estimate (95% Confidence Interval)p valueMaternal Loss in Years 2-41.79 (1.06-3.04)0.029Maternal Age at Offspring Birth0.98 (0.93-1.05)0.43

Model output of a mixed effects model of offspring survival to age 2, where mothers did not experience early maternal loss or the presence of a close in age sibling:

ParameterHazard Ratio Estimate (95% Confidence Interval)p valueMaternal Loss in Years 2-41.28 (0.7-2.34)0.43Maternal Age at Offspring Birth0.98 (0.92-1.04)0.43

Finally, to address the possible role of maternal age in offspring survival more broadly, we have built alternative models to the model presented in the main text that additionally include fixed effects of maternal age or maternal age and its interaction with maternal early adversity. Neither maternal age or its interaction with adversity were significant predictors in the models, a result that we reference in the Results section and include in Supplementary file 4 and Supplementary file 5.

2) The logic of why short inter-birth interval is used to as a measure early maternal adversity, but not offspring adversity, is not clear to me.

We inadvertently made this point more complicated than it is. Because females who lose their dependent young often quickly become pregnant again, offspring with the closest-in-age siblings are offspring who themselves died young. In such cases, the close-in-age siblings often occurred *as a result of* the focal offspring’s death, not as a *cause* of the focal offspring’s death. Thus, including short interbirth interval as a source of offspring adversity would make our results uninterpretable. We have attempted to clarify this issue in the Results section.

3) I do not find the analysis with the 10-minute focal follows (subsection “Maternal Early Life Adversity and Quantity of Maternal Care”) to be very helpful, as it seems unlikely that much useful information about the hypothesis can be gleaned from that short of a time. Using such data to assess how early adversity could shape maternal care is, however, a good idea, and instead of presenting this analysis I would recommend instead including a paragraph about this hypothesis/suggested method in the Discussion section as an important area of future research.

We agree that this analysis does not provide the reader with sufficient information to justify its inclusion, and so have dropped it from the manuscript. Instead, as the reviewer suggested, we have added a paragraph to the Discussion section in which we highlight the hypothesis and outline the types of data that would be necessary to test it.

4) In results the authors critique inconsistent sex-specific effects in human literature (Discussion section) – but do the authors look for sex-specific effects in their analysis? Given prior research that suggests sex-differences in sensitivity to early environmental experience this seems like something important to assess.

In response to the reviewer’s suggestion, we have added an analysis to the supplementary materials that considers offspring sex and interactions between offspring sex and maternal early adversity as fixed effects. These models do not perform better than the model reported in the main text. We reference this supplemental analysis in the Results section. We cannot differentiate between an explanation in which sex differences do not occur and one in which interaction effects are too small to detect in our analysis, due to power. We have therefore elected to leave sex-specific analyses out in the main text.

5) I would have appreciated seeing more of a discussion of why the authors believe the variables that came out as significant – in particular maternal death and a short interbirth interval – are most strongly associated with offspring survival.

We thank the reviewer for this helpful suggestion. In the Discussion section we have added several sentences on this topic, linked to our discussion of maternal care.

Reviewer #3:This is an important study demonstrating a likely intergenerational maternal effect on juvenile survival among baboons at Amboseli, one of the longest running studies of its kind. The authors use multiple decades of data and a large sample to report evidence that a female's early life stress experiences can influence survival of their unexposed juvenile offspring, independent of their own early life experiences. This is an important finding that extends study of multi-generational parental effects to a long-lived mammal/primate. It also provides evidence against the applicability of the predictive adaptive response hypothesis in this setting. Overall, the paper is nicely written and clear, and I have relatively minor suggestions on how to improve it.The biggest issue that I see is in the set up and framing with respect to past literatures – the importance of the study is framed with respect to the human chronic disease-motivated DOHaD literature, which seeks to understand the developmental contributions to late life outcomes like diabetes or CVD. This study, in contrast, focuses on juvenile mortality. The authors do not provide adequate information on the most typical causes of juvenile death. And in any case, it is hard to see juvenile mortality (which I assume ultimately traces back to factors like resource access, social capital, violence?) as a model or analogue for the types of pathways and outcomes studied by DOHaD.Which makes me wonder whether it might be more appropriate, and more to the point, to frame the primary contribution and motivation of this paper with respect to the parental effects literature, rather than DOHaD. I see little similarity (biologically) between what is being studied here and what the DOHaD literature studies, so it feels a bit distracting to have DOHaD as the opening framing – especially given that the authors never come back to that literature or discuss the limited insights that these findings provide into it. Another possible, generic framing is a test of the PAR idea – these findings showing pretty clear evidence against its applicability in this case.

We thank the reviewer for these helpful comments, which have caused us to edit our text in several ways. First, we agree that our inclusion of a list of morbidities and sources of mortality in the first paragraph of the introduction might lead readers to think that we would present a comparable analysis of causes of death. To eliminate this problem, we have re-written this statement in the Introduction as follows, “For example, in humans, low early life socioeconomic status predicts increased increased risk of mortality and many measures of poor health (Beebe-Dimmer et al., 2004; Frankel et al., 1999; Galobardes et al., 2004; Kittleson et al., 2006; Kuh et al., 2002; Lidfeldt et al., 2007; Naess et al., 2004; Smith et al., 1998; Van De Mheen et al., 1998).”

We have also substantially reduced the scope of our discussion by arguing only that our study should inspire similar studies in humans, instead of attempt to directly link our findings to human health.

We view our study as connected to an extended version of the DOHaD hypothesis that seeks to quantify the persistence of developmental perturbations over time, including in subsequent generations. However, we agree that our analysis also fits into the literature on parental effects, and we have reframed our introduction accordingly (Introduction).

These modifications in our text are meant to remove our reliance on the DOHaD literature for framing.